# The Integration of IoT (Internet of Things) Sensors and Location-Based Services for Water Quality Monitoring: A Systematic Literature Review

**DOI:** 10.3390/s25061918

**Published:** 2025-03-19

**Authors:** Rajapaksha Mudiyanselage Prasad Niroshan Sanjaya Bandara, Amila Buddhika Jayasignhe, Günther Retscher

**Affiliations:** 1Department of Town and Country Planning, University of Moratuwa, Moratuwa 10400, Sri Lanka; niroshans@uom.lk (R.M.P.N.S.B.); amilabj@uom.lk (A.B.J.); 2Department of Geodesy and Geoinformation, TU Wien—Vienna University of Technology, 1040 Vienna, Austria

**Keywords:** IoT sensors, LBS, water quality (WQ) monitoring, current systems and applications, novel developments

## Abstract

The increasing demand for clean and reliable water resources, coupled with the growing threat of water pollution, has made real-time water quality (WQ) monitoring and assessment a critical priority in many urban areas. Urban environments encounter substantial challenges in maintaining WQ, driven by factors such as rapid population growth, industrial expansion, and the impacts of climate change. Effective real-time WQ monitoring is essential for safeguarding public health, promoting environmental sustainability, and ensuring adherence to regulatory standards. The rapid advancement of Internet of Things (IoT) sensor technologies and smartphone applications presents an opportunity to develop integrated platforms for real-time WQ assessment. Advances in the IoT provide a transformative solution for WQ monitoring, revolutionizing the way we assess and manage our water resources. Moreover, recent developments in Location-Based Services (LBSs) and Global Navigation Satellite Systems (GNSSs) have significantly enhanced the accessibility and accuracy of location information. With the proliferation of GNSS services, such as GPS, GLONASS, Galileo, and BeiDou, users now have access to a diverse range of location data that are more precise and reliable than ever before. These advancements have made it easier to integrate location information into various applications, from urban planning and disaster management to environmental monitoring and transportation. The availability of multi-GNSS support allows for improved satellite coverage and reduces the potential for signal loss in urban environments or densely built environments. To harness this potential and to enable the seamless integration of the IoT and LBSs for sustainable WQ monitoring, a systematic literature review was conducted to determine past trends and future opportunities. This research aimed to review the limitations of traditional monitoring systems while fostering an understanding of the positioning capabilities of LBSs in environmental monitoring for sustainable urban development. The review highlights both the advancements and challenges in using the IoT and LBSs for real-time WQ monitoring, offering critical insights into the current state of the technology and its potential for future development. There is a pressing need for an integrated, real-time WQ monitoring system that is cost-effective and accessible. Such a system should leverage IoT sensor networks and LBSs to provide continuous monitoring, immediate feedback, and spatially dynamic insights, empowering stakeholders to address WQ issues collaboratively and efficiently.

## 1. Introduction

The increasing demand for clean and reliable water resources, coupled with the growing threat of water pollution, has made WQ monitoring and assessment a critical priority ([1,2]). Nowadays, environmental pollution causes substantial challenges in maintaining WQ, driven by factors such as rapid population growth, industrial expansion, and the impacts of climate change. Different types of water bodies are polluted due to various factors and sources of contamination. Many urban freshwater bodies are polluted due to point and non-point sources of pollution. The main reasons can be identified as sediments, pathogens, and nutrients contributed by a variety of sources, including stormwater runoff, leaking wastewater infrastructure, and channel erosion [3]. These create an adverse impact on surrounding water bodies during storm events when pollutants from urban areas are transported to downstream water bodies within a short period. Moreover, urban water bodies face significant ecological challenges, including weak circulation and vulnerability to pollution, largely influenced by their surrounding urban environments [4]. Effective management strategies, such as utilizing advanced data analysis techniques, are essential to mitigate these challenges and enhance water quality assessment. This phenomenon not only affects the ecological environment but also increases the risks to public health. Therefore, effective real-time WQ monitoring is essential for safeguarding public health, promoting environmental sustainability, and ensuring adherence to regulatory standards ([5,6]). Nevertheless, many existing monitoring systems are hindered by limitations in integration, accessibility, and scalability, reducing their effectiveness in addressing complex urban water management needs [7].

Traditional methods of environmental monitoring have long been constrained by limited temporal and spatial resolution, often resulting in delayed data acquisition and restricting the scope of data collection to smaller areas ([8,9]). These approaches are also labor-intensive, requiring manual sample collection followed by laboratory analysis. While electronic devices, such as handheld instruments for measuring parameters like the pH (Potential of Hydrogen), ORP (Oxidation–Reduction Potential), and conductivity, have been introduced, they primarily record data alongside sample identifiers, necessitating additional manual documentation [9]. Furthermore, these devices still operate on a single-sample basis, making continuous monitoring particularly challenging, especially over extensive areas. This approach demands substantial human intervention, with the temporal resolution remaining low since each location must be sampled individually [8]. Additionally, in traditional workflows, sampling locations are often documented on printed maps or manually recorded with associated coordinates and sample numbers, adding further complexity to data processing and analysis [10]. To address these limitations, emerging technologies such as the Internet of Things (IoT) and wireless sensor networks offer promising solutions for comprehensive, real-time environmental monitoring. The development of miniaturized, robust, and reliable sensors that can be deployed in pervasive networks has the potential to revolutionize the field of environmental monitoring ([11,12]). These advancements can enable continuous, high-resolution data collection across larger spatial scales, providing a more complete and dynamic understanding of environmental conditions.

The rapid advancement of IoT sensor technologies presents an opportunity to develop integrated platforms for real-time WQ assessment. Advances in the IoT provide a transformative solution for WQ monitoring, revolutionizing the way we assess and manage our water resources.

The IoT enables the deployment of low-cost, real-time sensors that continuously collect data on a wide range of water quality parameters, including the pH, temperature, dissolved oxygen, turbidity, conductivity, etc. These sensors can be strategically installed at specific locations and configured to measure the WQ parameters based on the conditions of those environments, both indoors and outdoors. Garcia Baigorri et al. [13] present a study on the role of smart sensors in enhancing urban water distribution systems, with a focus on sustainable city development. The study explored the integration of sensor technology to monitor the water quality and distribution efficiency. Further, IoT systems facilitate real-time monitoring, providing immediate access to data that can be transmitted to centralized platforms for visualization and analysis. This capability not only enhances the efficiency of data collection but also allows for rapid decision-making in response to emerging WQ issues [14]. Furthermore, the integration of IoT technology reduces the need for extensive human intervention, thereby minimizing potential errors associated with manual data logging and increasing the overall reliability of WQ assessments. By harnessing the power of the IoT, WQ monitoring can become more proactive and responsive, addressing the critical challenges posed by urbanization, industrialization, and climate change while promoting sustainable water management practices.

However, the need for reconfiguration procedures including initial installation costs and complex calibration can be identified as a limitation for fast deployment and easy relocation when collecting location-specific data [15]. If there is a need to relocate these sensors to a different site, they must be reconfigured to ensure accurate monitoring and data collection, which can slow down the overall process and limits their popularity. The requirement for reconfiguration when relocating sensors can hinder rapid data collection in different areas, potentially impacting the overall effectiveness of monitoring efforts. Thus, while the IoT offers significant improvements, these challenges must be considered to optimize its application in WQ management by leveraging Location-Based Services (LBSs) and Global Navigation Satellite Systems (GNSSs).

Recent developments in LBSs and GNSSs have significantly enhanced the accessibility and accuracy of location information. With the proliferation of GNSS services, such as GPS, GLONASS, Galileo, and BeiDou, users now have access to a diverse range of location data that are more precise and reliable than ever before [16]. These advancements have made it easier to integrate location information into various applications, from urban planning and disaster management to environmental monitoring and transportation. The availability of multi-GNSS support allows for improved satellite coverage and reduces the potential for signal loss in urban environments or densely built environments. This increased accessibility enables real-time tracking and monitoring, facilitating quicker decision-making processes in various fields. Additionally, the integration of GNSSs with IoT technologies and advanced sensors has streamlined data collection and analysis, enabling users to obtain accurate location information with minimal effort ([17,18]). Moreover, enhanced LBS applications now provide valuable insights into user behavior, traffic patterns, and resource management, contributing to smarter cities and more efficient systems. With the continued evolution of GNSS and LBS technologies, the use of location information has become far easier, empowering individuals and organizations to leverage spatial data for improved planning, resource allocation, and overall decision-making [19]. The integration of IoT sensor networks with LBS technologies and its transformative impact on positioning capabilities are utilized to enhance applications in environmental monitoring.

With the evolution of technologies, there are many recent review papers that have identified real-time IoT-based monitoring applications across various areas such as healthcare, infrastructure, industrial uses, and environmental and smart cities [20]. Also, review articles have focused on the deployment of IoT-based systems for WQ monitoring [21] and reviewed low-cost WQ monitoring sensors for integration with the IoT [22]. Further, there have been studies focused on advances in data analysis with the IoT and machine learning (ML) [2] and focused on water quality analysis with GIS and ML models [23]. Previous studies have primarily focused on various aspects of IoT-based WQ monitoring, including the deployment of IoT systems, the development of low-cost sensors ([21,22]), advancements in data analytics using the IoT and ML [2], and WQ analysis integrating GIS and ML models. However, these studies largely overlook the spatial context that LBSs provide. The integration of the IoT with LBSs remains an underexplored area, despite its potential to enhance real-time monitoring, improve geospatial data analysis, and enable location-specific decision-making for more effective water management.

The integration of the IoT and LBSs has been widely explored in various domains, including environmental monitoring, resource management, and security. However, the synergy between these technologies for WQ monitoring remains an emerging field. Several systematic reviews from the recent literature provide valuable insights into methodological advancements and research gaps. Recent reviews, such as those by Pettorru et al. [24], focus on localization techniques within IoT networks, emphasizing threats and mitigation strategies. This aspect is crucial for ensuring reliability in LBS applications, particularly in real-time environmental monitoring. Similarly, Kossieris et al. [25] discuss satellite-based inland water monitoring, highlighting the potential for integrating LBSs with remote sensing to enhance WQ assessment. While both studies contribute to understanding localization accuracy and large-scale monitoring, they do not explore the challenges of integrating IoT-enabled sensors with geospatial platforms.

In parallel, Alkhayyal and Mostafa [26] provide a systematic review on optimizing energy efficiency in LoRaWAN networks, a key aspect for scalable IoT applications. Their study underscores the role of AI and ML in improving data transmission and reducing energy consumption, which is highly relevant for IoT-based water monitoring systems. However, while energy optimization is well studied in general IoT applications, its specific impact on WQ monitoring remains underexplored. Additionally, Benicio et al. [27] review advanced modeling frameworks, such as the CE-QUAL-W2 model for reservoir eutrophication, illustrating how IoT data streams can enhance predictive accuracy. Although their study showcases the advantages of data-driven environmental modeling, it does not explicitly address how real-time geospatial tracking through LBSs could improve predictive capabilities.

The security challenges associated with IoT deployments have also been extensively examined. Sebestyen at al. [28] categorize vulnerabilities in IoT systems, noting that many LBS frameworks fail to incorporate adequate security measures. This gap is particularly relevant for WQ monitoring, where reliable data transmission is crucial for decision-making in public health and environmental management. Similarly, Lopez-Munoz et al. [29] present a wireless dynamic sensor network for WQ monitoring, demonstrating the IoT’s role in real-time data collection. Their study highlights how the IoT can improve responsiveness in environmental monitoring but does not delve into the interoperability challenges when integrating location-based tracking.

While several studies, such as those by Lukasik et al. [30], have explored AI-driven localization improvements, and Zamnuri et al. [31] investigate IoT-based aquaponics efficiency, few studies have explicitly linked these developments to comprehensive IoT-LBS integration for WQ assessment. This suggests a gap in research that fully leverages geospatial intelligence in real-time IoT applications. Moreover, the scalability of IoT-based monitoring remains a challenge, as discussed by Staniszewski et al. [32], who identify pressing issues in inland water research and monitoring. Addressing these gaps requires further exploration of how the IoT and LBSs can be jointly optimized to enhance WQ monitoring, ensuring real-time, secure, and scalable solutions that bridge the existing methodological and technological limitations.

This paper fills the gaps by exploring how the integration of the IoT and LBSs can enhance real-time, location-specific monitoring, enabling more precise decision-making and water resource management. While past studies have demonstrated the transformative potential of the IoT in enabling the large-scale implementation of environmental and WQ monitoring systems through the evolution of low-cost sensors and innovative wireless data transfer technologies [33], this research introduces an advancement by integrating LBSs into the IoT framework as an innovative solution to improve the limitations in the spatial and temporal resolution of the data. Moreover, this paper fills this critical gap by exploring how the integration of the IoT and LBSs can enhance real-time, location-specific WQ monitoring, enabling more precise decision-making and resource management. By systematically reviewing the existing literature, this study highlights the transformative potential of IoT-LBS integration in applications such as contamination tracking, predictive modeling, and optimized resource allocation, which neither technology can achieve independently. This paper systematically reviews the existing literature on the integration of the IoT and LBSs in WQ monitoring, synthesizing key findings from recent years. The primary objective is to analyze how these technologies enhance monitoring precision, facilitate informed decision-making, and contribute to sustainability. The review examines innovations in the integration of the IoT and LBSs within WQ monitoring systems by synthesizing significant insights and identifying advancements in research.

The remainder of this paper is structured as follows: Section 2 introduces and evaluates the methodology of the conducted literature review. Section 3, the core of the paper, presents and discusses the major findings of the study, addressing key questions raised through the systematic literature review. Finally, Section 4 provides the conclusions.

## 2. The Methodology of the Literature Review

### 2.1. Review Methodology

This study employed a systematic literature review (SLR) to explore the integration of the IoT and LBSs for WQ monitoring. This approach allowed for a comprehensive examination of relevant research, providing in-depth insights into how these technologies enhance monitoring precision and decision-making. The review outlines key strategies and methodologies, offering a clear understanding of how the IoT and LBSs are applied in WQ management and to what extent these technologies enhance the efficiency of WQ monitoring and provide support for decision-making. Additionally, the SLR highlights common challenges and identifies gaps in research, including areas that have been underexplored by researchers. By adhering to the Preferred Reporting Items for Systematic Reviews and Meta-Analyses (PRISMA) framework, this review offers a structured analysis, drawing from major databases such as the Web of Science (WoS) and Scopus to query and capture a broad range of relevant publications. The findings emphasize the role of the IoT and LBSs in improving water quality monitoring systems while promoting sustainability. The PRISMA framework comprises four primary components: (1) establishing eligibility criteria; (2) selecting from the acquired articles; (3) conducting data extraction; and (4) assessing the article quality to mitigate the bias risk and performing data synthesis. Also, defining eligibility criteria provides a focused review [34]. Systematic article selection helps to reduce subjectivity [35], while rigorous data extraction and quality assessment enhance reliability, contributing to a robust data synthesis process.

The initial stage entails establishing eligibility criteria through the identification of characteristics from previous research, utilizing the Participants, Intervention, Comparison, and Outcome (PICO) framework. The application of appropriate PICO criteria aids in the identification of specific research questions. The second stage involves the selection from the acquired articles, which can be achieved by identifying the number of independent reviewers at each phase. The third stage emphasizes the extraction of data regarding various factors, including the demographics, research methodology, and scope. In the third stage, analyses are conducted to assess the quality of articles to reduce the potential for bias. This stage employs the identification method to evaluate the risk associated with bias from prior research. The concluding phase entails synthesizing data derived from the outcomes of quantitative analysis. Figure 1 illustrates the stages of the PRISMA.

### 2.2. Search Strategy (Eligibility Criteria Determination)

To conduct an SLR on the integration of the IoT and LBSs for WQ monitoring, a comprehensive search strategy was employed to identify relevant studies. The databases targeted for the review were the WoS and Scopus, which are recognized for their extensive academic resources and credibility. Searches on both platforms were carried out separately to follow a uniform process. This helped to minimize the bias in the search and query steps.

The search queries were decided by including keywords such as “Internet of Things (IoT)” and “Location-Based Services (LBS)”. These were initially selected based on what was found in the title, abstract, and keywords of the manuscripts to identify past studies and explore the integration of these technologies and its trend in previous decades. These keywords were chosen to uncover research that examined how the IoT and LBSs have been combined and applied across various fields over time. The goal was to identify research on the recent development and integration of these technologies, providing insights into their integration capabilities and how they have evolved in modern applications. After that, we narrowed our search down specifically to the context of WQ monitoring. During our study, we found research studies that did not use the term “LBS” but did use global positioning or GPS, and only a few research papers used the term Global Navigational Satellite Systems or GNSSs to develop their applications for location information integration.

This study used Boolean operators like AND and OR systematically to refine the search and ensure more precise results. This specific search aimed to identify research that concentrated solely on how IoT and LBS technologies are being applied to enhance WQ monitoring. The refined search criteria were as follows:

((“Internet of Things” OR “IoT”) AND (“Location-Based Services” OR “LBS” OR “Global Navigational Satellite Systems” OR “GNSS” OR “Global Positioning System” OR “GPS”) AND (“Water Quality” OR “Water Management”)).

This Boolean search query was designed for use in academic databases such as Scopus and the WoS to retrieve relevant research on the integration of IoT and location-based technologies for WQ monitoring and management. The query consisted of three main components: (1) “Internet of Things” OR “IoT”, ensuring the inclusion of studies related to IoT applications; (2) “Location-Based Services” OR “LBS” OR “Global Navigational Satellite Systems” OR “GNSS” OR “Global Positioning System” OR “GPS”, which captured various location-based technologies essential for geospatial monitoring; and (3) “Water Quality” OR “Water Management”, ensuring the search retrieved studies focusing on water monitoring and resource management. The AND operator ensured that the results included at least one term from each category, while the OR operator broadened the search to capture different terminology used in the literature. This structured approach maximizes the retrieval of relevant studies while filtering out unrelated research, making it particularly useful for systematic literature reviews (SLRs) and bibliometric analyses in the fields of environmental monitoring, IoT applications, and geospatial technologies.

### 2.3. Screening and Eligibility

After deciding on the search strategy, inclusion and exclusion criteria were determined before conducting the searches. A structured methodology section for a systematic review outlines the inclusion and exclusion criteria, ensuring a robust selection process. The inclusion criteria focused on peer-reviewed articles published in specific journals, while the exclusion criteria ruled out non-English publications or studies lacking relevant data. These criteria enabled us to filter the publications, to ensure the relevance and quality of the studies selected for this systematic literature review, and to specific the inclusion and exclusion criteria that would apply during the search and screening process. Prior to the screening procedure, inclusion and exclusion criteria were established to ensure transparency and reliability, as well as to ensure that only the most relevant studies were selected for screening.

#### 2.3.1. Inclusion Criteria

The following inclusion criteria were applied:
The selected studies had to specifically address the integration of the IoT and LBSs in the context of WQ monitoring or water management.Only peer-reviewed journal articles were considered to maintain a high level of methodological accuracy, ensuring the high reliability and validity of the research papers selected for the review.The articles needed to be written in English.The studies needed to provide empirical data, detailed case studies, or comprehensive theoretical analyses that contributed directly to understanding the integration of the IoT and LBSs.Given the rapid evolution of IoT and LBS technologies, studies published within recent years were prioritized to ensure the review reflected current technological standards.

#### 2.3.2. Exclusion Criteria

The following exclusion criteria were applied:
Articles that were not written in English were excluded.Conference papers, theses, reviews, or any non-peer-reviewed publications were excluded to ensure robust quality control.Duplicate articles identified during the initial search were removed.Studies that discussed the IoT or LBSs in unrelated fields such as transportation, healthcare, or urban planning without direct application to WQ monitoring were excluded.Studies that discussed the IoT and LBSs separately, without addressing their integration for WQ monitoring, were not included.

### 2.4. Article Selection

The results of the literature search were recorded in an Excel spreadsheet. Then, the precision of the search keywords, along with the inclusion and exclusion criteria, was assessed. Additionally, the researcher conducted a simultaneous and independent screening of titles and abstracts according to the established inclusion criteria.

The article selection stage sought to identify all articles that fulfilled the criteria for inclusion in the systematic literature review process. The selection criteria were implemented in two phases: the initial phase assessed the title and abstract concurrently, followed by a second phase that evaluated the full text.

#### 2.4.1. Title and Abstract Screening

In the screening process, articles that did not meet the inclusion criteria were excluded. In cases where the relevance was uncertain based on the title and abstract alone, the full text was retrieved for further evaluation. The key questions addressed were the following:
In what type of environment was the system being implemented?What sensors were used and which parameters were measured to monitor WQ?What methods were employed to collect location information?What was the spatial and temporal resolution in the data collection and transfer method?How do these studies demonstrate improvements through IoT and LBS integration?How do the selected studies highlight the impact of IoT and LBS technologies on decision-making in water resource management?

By addressing these questions, this review aimed to synthesize key insights, technological advancements and integration, and areas requiring further exploration within the selected articles.

#### 2.4.2. Full-Text Screening

The full texts of the remaining studies were reviewed in detail against the inclusion and exclusion criteria. Key characteristics (such as the study’s approach, methodology, and findings related to the integration of the IoT and LBSs) were systematically extracted.

A PRISMA flow diagram (Figure 2) is provided to visually represent the screening process, detailing the number of studies identified, screened, excluded, and ultimately included in the review.

#### 2.4.3. Data Extraction and Synthesis

The data extraction phase was initiated with the collection of data on articles that fulfilled the criteria for full-text selection. The data synthesis stage commenced with the integration of findings derived from the quantitative analysis conducted on the article quality. The extraction results from data items important to the research question were synthesized to create a comprehensive framework of the integration of the IoT and LBSs for WQ monitoring. A content analysis was performed for this purpose.

## 3. Analyses of Major Findings

### 3.1. Data Synthesis

The search of the databases resulted in the identification of a total of 89 articles published between 2000 and 2024. The identified articles were analyzed emphasizing the trend and pattern of publications in each database before applying the inclusion and exclusion criteria and answering the study questions.

The bibliometric analysis of Scopus and WoS publications showed a growing interest in research related to the IoT, LBSs, and WQ over time between 2000 and 2024. As is shown in Figure 3, the trend indicates a steady increase in publications across both databases, reflecting the expanding role of these technologies in water quality monitoring. The combined cumulative trend from Scopus and the WoS confirms a continuous rise in the scholarly output, with notable peaks after 2020.

Between 2000 and 2024, technological advancements profoundly transformed various sectors. Wang et al. [36] highlight that the evolution of the IoT can be identified as taking place in three different stages over the time period: the early stages (2002–2009)—the emergence of the IoT as a research area; expansion (2009–2015)—the development period; and maturity (2015–2019)—the rapid growth in IoT applications and maturation of IoT technologies. The same pattern is reflected throughout this analysis across these decades. The deployment of 5G technology since 2020 has significantly enhanced IoT connectivity, enabling more efficient communication and interactions among devices. This advancement has driven societal change, particularly in sustainability efforts, as noted by [37], illustrating the crucial role of societal acceptance in leveraging 5G for transformative development. The deployment of 5G networks became a significant driver for the IoT research area [38]. The emergence of the IoT impacted massively on digital transformation across numerous sectors [39]. When it comes to the integration of LBSs and the IoT for water quality monitoring, the same pattern can be observed.

Among these studies, Figure 4 reveals that the IoT was the most frequently mentioned term. This highlights that the IoT plays the central role in technological advancements. Research on monitoring and WQ indicated the use of IoT applications, which shows a strong focus on environmental and water-related studies. The term unmanned surface vehicle (USV) appeared as the fourth most frequently used word, and it reflects the interest in autonomous systems for monitoring and data collection. Sensors being the next most significant word among the other keywords emphasizes the growing use of sensor technology for real-time data acquisition. Other keywords, including GPS, tracking, and sustainability, showcase the interest in geospatial and environmental sustainability studies. The importance of fundamental water parameters like the pH, temperature, conductivity, and dissolved oxygen in WQ assessment was underlined. Additionally, the mention of machine learning shows an emerging trend in AI-driven environmental analysis. Other relevant keywords, such as low-cost sensors, indicate growing interest in cost-effective and scalable monitoring solutions.

These identified keywords were categorized into groups such as the IoT, LBSs, and WQ. Following this, the frequency of these grouped keywords was analyzed over different publication years, as shown in Figure 5. This revealed emerging trends and shifts in the research focus. The visualization of these trends using a time series plot highlights the increasing amount of research on the IoT, LBSs, and WQ. It gives us a better understanding of technological advancements and developments in the IoT, geospatial technologies, and water quality monitoring.

Since 2000, research on IoT sensors has shown a steady upward trend, particularly accelerating towards 2020, marking the transition from an early exploratory phase to a more developed stage. During this period, LBSs remained largely unexplored. However, with advancements in technology, the simultaneous growth of the use of both the IoT and LBSs in WQ monitoring became evident around 2020. The latest trend indicates that while the IoT has reached the maturity stage, the application of LBSs is in its development phase, emphasizing the growing need for integrating LBSs with the IoT to enhance WQ monitoring and management.

Moreover, a co-occurrence analysis was performed by using VOSviewer, which is a tool for network-based visualization that helps map relationships between keywords, authors, citations, and co-authorship in research publications. VOSviewer (Version 1.6.20) was applied for co-occurrence analysis, along with the Scopus and WoS databases. The terms IoT, GPS, WQ, and WQ monitoring systems formed a central cluster, with sub-networks linking to related terms such as unmanned surface vehicle, internet, sensors, tracking, and real-time monitoring, as shown in Figure 6.

Next, a Python (version 3.9.4) script was used to analyze the keyword co-occurrence in Scopus and WoS data to visualize relationships between key research topics using NetworkX (version 3.2.1) and Matplotlib (version 3.9.4). It constructed a co-occurrence matrix, identifying frequently co-occurring keyword pairs across studies. A graph network was built where nodes represent keywords and edges indicate co-occurrences.

The keyword co-occurrence network shown in Figure 7 provides a visual representation of the interconnections between key terms in the domains of the IoT, GPS, GNSSs, LBSs, and WQ monitoring, based on the systematic literature review. The diagram highlights major research themes, with the IoT and WQ monitoring emerging as central topics, closely linked to real-time monitoring, environmental sensors, and machine learning. The presence of the GPS and GNSS domains underscores the significance of geospatial technologies in water monitoring applications, particularly in precision agriculture and remote sensing. Additionally, the mention of AI-driven techniques, such as ML and predictive modeling, demonstrates the increasing role of computational methods in data analysis and decision-making. The strong interconnectivity among keywords reflects the multidisciplinary nature of this research field, integrating the IoT, geospatial analysis, and environmental monitoring. Furthermore, the inclusion of emerging technologies like blockchains, cloud computing, and neural networks highlights evolving trends in data security, storage, and intelligent automation. This network effectively captures the breadth of research in this area, making it a valuable tool for identifying key topics and future directions.

Since 2020, significant advancements have been observed in IoT applications, particularly in the integration of LBSs for environmental monitoring, including WQ assessment. Prior to this period, IoT technology was in its developmental phase, primarily focusing on hardware advancements and the incorporation of sensors for environmental data collection. During this early stage, research primarily aimed to evaluate the applicability and usability of the IoT in WQ monitoring, leading to numerous systematic reviews and studies on sensor integration and low-cost sensor development.

As IoT technology matured, the focus shifted toward enhancing scalability, interoperability, and real-time analytics, paving the way for the integration of LBSs. This integration enables spatially aware IoT systems, allowing for precise geospatial data collection, improved resource allocation, and more effective responses to WQ issues [40]. By leveraging LBSs, IoT applications can provide location-specific insights, facilitating data-driven decision-making and empowering stakeholders with actionable information [41]. Additionally, LBS-enhanced IoT systems contribute to increased public awareness and proactive community engagement in water resource management [42].

Given that previous studies have extensively explored IoT sensor integration and cost-effective sensor development, this study distinguished itself by investigating the period of IoT maturation, specifically focusing on its integration with LBSs. By examining this emerging synergy, the study aimed to identify novel contributions, address existing research gaps, and highlight future directions for sustainable WQ monitoring [43,44].

By considering the stages of IoT and technological advancement as described in this paper and identifying trends and patterns in the rapid growth in IoT applications, the maturation of IoT technologies, which has occurred from 2020 to the present, was selected for further investigation. Since this paper is reviewing the recent advancement in IoT and LBS integration, studies were selected from 1 January 2020 to 31 December 2024 (5 years).

As a result of this focused search, a total of 77 relevant studies were identified from Scopus (56) and the WoS (21). This indicates a limited but specific body of literature that directly addresses recent developments in the integration of the IoT and LBSs in the context of WQ monitoring. Despite the smaller number of studies, these studies are significant as they reflect the targeted research efforts in this emerging field, offering insights into current applications and future directions for technological advancements in water management.

After selecting those studies, inclusion and exclusion criteria were applied for the filtering. Then, 16 papers were selected from the WoS and 12 from Scopus as the initial pool of 28 studies. After that, nine articles were excluded as they appeared as duplicates in both databases, and one was not in the scope of the study. Another two articles were excluded due to the language constraints, and those were not written in English. The systematic review protocol is shown in Figure 2. The 16 selected studies provide valuable insights into the integration of the IoT and LBSs in WQ monitoring, highlighting technological advancements and identifying research gaps that will guide future investigations in this field.

### 3.2. Answering the Questions

The following sections present the key findings from the review and address the questions outlined in this paper.

#### 3.2.1. In What Type of Environment Was the System Being Implemented?

The systems in the studies described were being implemented in a diverse range of aquatic environments, each with unique monitoring requirements. The studies mainly included freshwater bodies such as rivers, lakes, and urban reservoirs for WQ assessment and pollution detection. In marine and coastal environments, such as seashores, ports, and aquaculture systems, they focused on monitoring salinity and sea surface temperature change detection. Water treatment plants utilize WQ monitoring systems to ensure the quality of processed water. Aqua farming and fishponds are focusing on maintaining optimal conditions for aquatic life. Additionally, the systems are deployed in remote and challenging environments, such as alpine lakes, where they operate with minimal logistical support. Laboratory testing has also been conducted to validate sensor performance and test the accuracy of the developed systems. Some systems are designed for dynamic platforms, such as boats, buoys, and even surfboard fins, enabling mobile monitoring and tracking. Public tap water systems and large-scale water grids have also been covered, with smartphone-based sensors and location-enabled networks providing real-time data. Figure 8 depicts the analysis of the types of environments in which the systems were applied; the findings highlight that the major applications are predominantly in large-scale water bodies, where location information plays a crucial role. GNSSs and UAV-based tracking methods were more commonly used for water monitoring in urban water bodies compared to being applied in rural, small-scale, and industrial water monitoring contexts.

Overall, these systems are designed to operate in both static and dynamic water body settings to address the specific needs of each environment, from flood warnings and pollution detection to aquaculture management and remote lake monitoring and seashore WQ monitoring.

#### 3.2.2. What Sensors Were Used and Which Parameters Were Measured to Monitor WQ?

It was important to identify the sensors which were used to measure the WQ and what parameters were measured. The identification of the sensors used and the number of parameters measured provided the level of robustness and the level of scalability for deploying the developed systems for different environments and water bodies.

WQ monitoring can be performed by using various sensors to assess whether the measured parameters are at a healthy level in aquatic environments. In the selected articles, common parameters most studies selected to measure water quality were the temperature and pH. The electric conductivity (EC) could be identified as a second common parameter. The salinity and Total Dissolved Solids (TDSs) were calculated using EC values [45]. Next, the dissolved oxygen (DO) and Oxidation–Reduction Potential (ORP) were the most common parameters in these studies. After carefully reviewing the articles, a Python (version 3.9.4) script was developed to extract and analyze the frequency of the WQ measurement parameters that were used across all the studies. The results are presented in the description above and illustrated in Figure 9.

These parameters were measured with sensor probes, mainly manufactured by DFRobot and Atlas Scientific. A CTD probe, an instrument used in oceanography, also referred to as an oceanographic sonde, was used to measure the electrical conductivity, temperature, and depth in Thong-Un et al. [45]. Along with that, the study used a bathymetry sensor as a dual-frequency echosounder for mapping the lakebed topography. pH, EC, and DO probes are sensitive devices for measuring values [44] and face rapid degradation [22] when exposed to the environment. Among the studies, Thong-Un et al. [45] and Kinar and Brinkmann [46] measured the turbidity, which measures the quantity of suspended particles by measuring how cloudy or hazy a liquid is. All these sensors are compatible with Arduino-like microcontrollers.

#### 3.2.3. What Methods Were Employed to Collect Location Information?

The identification of the geospatial location plays a vital role in WQ monitoring. It provides a meaning for the WQ parameter values. Therefore, this section reviews what methods were employed for collecting precise location information. Most studies highlighted the method of obtaining location information using a GPS module. Studies such as Amuthakkannan and Al Yaqoubi [47] Kinar and Brinkmann [46] and Zhang et al. [48] highlighted the method of obtaining location information using GNSS receivers. This shows that the majority of studies used the term global positioning or GPS as the term for location-based services.

Various methods were employed to collect location information in the reviewed studies, primarily utilizing GNSS constellation modules and GPS support modules alone. Among the commonly used modules, the u-blox Neo-6M ([3,47]) and UBlox-M8N [49] were widely adopted for several studies. The u-blox Neo-6M supports GPS, while the UBlox-M8N supports GPS, Galileo, GLONASS, and BeiDou. The BN-180 GPS unit [50] is capable of supporting GPS, GLONASS, Galileo, BeiDou, QZSS, and SBAS and is primarily used for LBS positioning. Similarly, the u-blox CAM-M8Q [51] functions as a GNSS module, also supporting GPS, Galileo, GLONASS, and BeiDou. Some studies employed LQBD1202 [52], which is a dual-frequency GNSS positioning module for Beidou and GPS. The SparkFun GPS-RTK2 Board (ZED-F9P) with an ANN-MB-00 antenna [53] offers enhanced real-time kinematic (RTK) positioning for improved precision. Additionally, communication-integrated modules such as the SIM808 [54], which combines GPS and GSM, were also used in some studies. The SparkFun NEO-M8U [45], with its sensor fusion capabilities, contributes to more robust and diverse location data collection approaches. These modules demonstrate the diverse range of technologies utilized for location-based data acquisition, varying from standard GNSS receivers to high-precision GNSS and RTK systems, depending on the specific requirements of each study. The specifications of the modules are shown in Table 1.

While the GNSS specifications were described in these studies, some studies [55] did not highlight the method or the module used for the systems and data collection. Also, for studies like Kimothi et al. [56] using GPS as a keyword, the location information should be collected separately. According to the author, after the site selection, a GPS survey should be carried out for the generation of pond locations.

#### 3.2.4. What Was the Spatial and Temporal Resolution in the Data Collection and Transfer Method?

The identified studies mainly discussed the hardware components and their uses and specifications, emphasizing the development of IoT-based WQ monitoring systems. Also, these studies mainly reported the data accuracy and applicability. They also mentioned the hardware components and each component’s function and duration of operation. Moreover, these studies emphasized the frequency or the temporal resolution, which could be adjusted according to the deployment. The results discussed in the papers were demonstrated in experimental situations. For example, Agade and Bean [3] showed that their survey lasted 37 h with a 5 min sampling frequency, and they deployed sensors at two trial WQ survey locations. Similarly, Kinar and Brinkmann [46] mentioned that the sampling interval could be set using the command line interface (CLI) as an additional functionality. During the laboratory testing, 365 samples were collected for approximately 31 h with 5 min sampling intervals. However, during the field deployment, samples were collected every 30 min, stored on an SD card and sent via a cellular network to a server. Moreover, Blanco-Gomez et al. [50] reported that the developed system used to measure the WQ every 3 s was sending and logging data on a smartphone for more than 56 h until the battery ran out. Additionally, it was deployed on a boyer and monitored the EC for more than 21 h. Also, Bresnahan et al. [51] mentioned that data were logged at 1 Hz on a fully charged device. This study also highlighted that the logging duration depended significantly on the battery, which had the capacity for roughly 10 h of continuous logging and transmission via a cellular network. Odetti et al. [53] also reported that their system could sustain continuous operation for up to 10 h. Moreover, it could extend its operational duration further with intelligent power management strategies. Staude et al. [54] also reported that their system allowed for the collection, processing, and analysis of data in real time or at a specified interval. However, data recording took place every 30 min and included the values of four measured parameters. From these studies, it can be seen that more of a focus on system development has been promoted and the applicability of the systems discussed. This is not sufficient to guarantee the location accuracy or the temporal and spatial resolution, since the studies have highlighted that the operation duration is very much dependent on the battery life-time.

The spatial and temporal resolution of the data collection and transfer methods varied depending on the application and the study. Also, the spatial and temporal resolution changed significantly based on the GNSS module used, data acquisition frequency, and communication interfaces. The spatial resolution refers to the positional difference and accuracy of the collected data. The accuracy can vary from a few meters based on the module (e.g., a u-blox Neo-6M with a ∼2.5 m CEP) to centimeter-level precision in RTK-enabled modules like the ZED-F9P (∼1 cm with RTK). The temporal resolution is determined by the data sampling rate, which typically ranges from once per second to 10 times per second in high-performance GNSS modules. High-temporal-resolution modules help to enhance tracking systems. The data transfer method also influences the resolution and efficiency of data collection. Modules like the SIM808 integrate GSM communication, enabling remote data transmission over cellular networks. Other GNSS-embedded systems require a separate communication module to transfer data. Thus, the choice of the GNSS module and transfer method is determined based on the spatial accuracy, update frequency, and real-time data transmission capabilities. Also, GNSS modules are subject to significant power consumption, which reduces the operational duration of the systems.

#### 3.2.5. How Do These Studies Demonstrate Improvements Through IoT and LBS Integration?

The integration of the IoT and LBSs in the studies demonstrates significant advancements in WQ monitoring systems. The IoT enables real-time data collection from sensors measuring parameters such as the pH, temperature, turbidity, and dissolved oxygen, while LBSs embedded with GNSS modules provide precise location tracking. Together, these technologies enhance the accuracy, efficiency, and scalability of monitoring systems across diverse environments. For instance, systems reported in studies such as Agade and Bean [3] and Blanco-Gomez et al. [50] transmit data to cloud servers, which enables remote monitoring and the quick detection of abnormalities and changes in WQ. This real-time capability is crucial for change detection and rapid responses.

The integration of LBSs improves spatial accuracy, allowing for the detailed mapping of WQ variations. For example, Zhang et al. [48] utilized a dual-frequency GNSS module to achieve precise positioning for a 3D WQ monitoring system, while Odetti et al. [53] employed RTK-GNSS for high-precision tracking with RTK in remote alpine lakes. This level of accuracy is essential for understanding spatial patterns and addressing localized issues in large or dynamic water bodies. Additionally, the IoT and LBSs enable systems to operate autonomously in remote locations, reducing the need for manual intervention and logistical support.

The IoT and LBSs also enhance decision-making through advanced data visualization and analysis. Systems like those developed by Odetti et al. [53] use cloud platforms and web applications to provide dashboards to visualize data as charts and maps, enabling stakeholders to understand them easily. This is particularly valuable for managing aqua farming, as demonstrated by Qadir et al. [55], where data were visualized on web maps to optimize fishpond conditions. Furthermore, the integration of the IoT and LBSs reduces costs and increases adaptability, making advanced monitoring systems accessible for a wide range of applications. For example, Blanco-Gomez et al. [50] and Baghel et al. [57] highlights low-cost solutions that leverage the IoT and GNSSs for scalable and effective monitoring.

Moreover, as shown in Figure 10, the reviewed studies highlight key improvements focusing on the accuracy of WQ monitoring sensors, sustainability, efficiency, and real-time monitoring, rather than emphasizing location precision.

Among the various improvements demonstrated through IoT and LBS integration, accuracy stands out as the most frequently emphasized factor in the reviewed studies, as shown in Figure 10. This high emphasis on accuracy reflects the critical need for precise and reliable WQ monitoring, particularly in dynamic and large-scale environments. The studies highlight the role of high-resolution sensors, GNSS-based positioning, and advanced data processing algorithms in minimizing measurement errors and enhancing the reliability of the collected data. For instance, Zhang et al. [48] illustrate how dual-frequency GNSS modules improve spatial positioning accuracy in WQ monitoring, allowing for the more precise mapping of pollution sources. Similarly, IoT-based multi-sensor networks, discussed in Kimothi et al. [56], ensure that key WQ parameters such as the pH, turbidity, and dissolved oxygen are recorded with minimal deviations, improving detection capabilities for subtle environmental changes. Enhanced accuracy not only facilitates real-time anomaly detection but also enables more effective decision-making for environmental management, regulatory compliance, and predictive modeling.

The combination of the IoT and LBSs also provides dynamic and mobile monitoring, which enables WQ monitoring for moving water bodies or floating platforms. Bresnahan et al. [51] used an oceanographic sensor-equipped surfboard fin to collect data in coastal regions, while Blanco-Gomez et al. [50] deployed buoys and boats for mobile monitoring in ports and aquatic environments. This flexibility plays a vital role in capturing temporal and spatial variations in WQ. Additionally, the IoT and LBSs enhance early warning systems and predictive capabilities, where a combination of the IoT, GNSSs, and GIS provides early warnings for floods and where water level and air quality monitoring support the use of flood and pollution alerts.

Finally, the IoT and LBSs enable citizen science initiatives, engaging the public in environmental monitoring for sustainable development. Amuthakkannan and Al Yaqoubi [47] demonstrates how smartphone-based sensors and GNSSs can empower individuals to contribute to WQ monitoring, enabling public participation. In summary, the integration of the IoT and LBSs in these studies showcases significant improvements in real-time data collection, spatial accuracy, remote operation, decision-making, cost-effectiveness, and public engagement, making WQ monitoring systems more efficient, usable, and important across diverse environments.

#### 3.2.6. How Do the Selected Studies Highlight the Impact of IoT and LBS Technologies on Decision-Making in Water Resource Management?

The selected studies demonstrate the transformative impact of IoT and LBS technologies on decision-making in water resource management. By enabling real-time data collection, IoT systems continuously monitor critical WQ parameters such as the pH, temperature, turbidity, and dissolved oxygen, providing stakeholders with immediate insights for timely action. This real-time capability is essential for minimizing environmental damage and providing quick responses to WQ changes. Meanwhile, LBSs, powered by GNSSs, enhance spatial accuracy, enabling the mapping of the WQ variations of large water bodies. Studies such as Zhang et al. [48] and Blanco-Gomez et al. [50] highlight how dual-frequency GNSS modules and RTK-GNSS provide high-precision location tracking, helping to identify changes in sensitive ecosystems like alpine lakes.

The integration of the IoT and LBSs also facilitates predictive analytics and early warning systems, which are critical for decision-making. Also, larger location-specific data sets provide the predictive capabilities to allow authorities to take preventive measures, reducing the impact of environmental hazards. Additionally, IoT platforms and LBSs enable advanced data visualization through dashboards, generating maps and charts, making complex data more accessible and understandable. Also, such dashboards present data in user-friendly formats, improving decision-making and supporting regulatory authorities.

Moreover, IoT and LBS technologies reduce the cost of WQ monitoring, making advanced systems accessible for a wide range of applications. These cost-effective systems also enable remote and autonomous monitoring, as highlighted in the word cloud analysis given in Figure 11. Finally, the IoT and LBSs foster public engagement and enable citizen science, where smartphone-based sensors and GNSSs empower individuals to contribute to WQ monitoring. By enhancing real-time data collection, spatial accuracy, predictive capabilities, and public involvement, IoT and LBS technologies significantly improve decision-making in water resource management, leading to a more sustainable future.

## 4. Conclusions

The integration of IoT and LBS technologies for WQ monitoring offers substantial improvements in both spatial and temporal resolution, transforming water resource management from focusing on static to dynamic water environments. IoT sensors enable the continuous, real-time data collection of WQ parameters such as the temperature, pH, and dissolved oxygen, allowing for high-frequency monitoring that enhances the temporal resolution of water monitoring. Also, they provide easy customization and changes to parameters. By capturing data consistently over time, stakeholders can quickly identify changes, track trends, and respond to potential threats. Additionally, the use of autonomous platforms and sensors ensures comprehensive data collection across diverse environments, from urban reservoirs to remote lakes and rivers. LBS technologies further enhance the spatial resolution by providing precise georeferencing through tools like RTK-GNSS and geotagged sensor data. These technologies enable the accurate mapping of WQ conditions, allowing decision-makers to identify specific areas of concern, such as pollution hotspots, that spatially impact on surrounding water resources or vulnerable ecosystems.

While this review covers the relevant literature on IoT and LBS integration for WQ monitoring, it also provides a clear comparison to existing surveys, emphasizing its unique contributions. However, information is lacking on the types of GNSS receivers used, their specifications, accuracy, precision, GNSS signal characteristics, positioning accuracy and performance, latency, calibration procedure, and reliability for specific applications. Most of the work used GNSSs for autonomous system development, but none of the work discussed the limitations regarding the latency, errors, and limitations of LBSs. Previous reviews have focused on IoT applications in WQ monitoring [21], low-cost sensors [22], and data analytics using the IoT and ML [2], but they often overlook the impact of the spatial context that LBSs provide.

Additionally, most studies discussed more critical aspects of limitations, biases, and unexplored areas in previous work. For instance, while existing studies have made significant progress in IoT-based WQ monitoring, they often lack a focus on scalability in remote areas, cost-effectiveness, and standardized frameworks for IoT-LBS integration.

The combination of high temporal and spatial resolution through IoT and LBS integration supports data-driven decision-making, leading to more informed, efficient, and sustainable water resource management practices. The evolution of these technologies from basic IoT systems to advanced platforms and RTK-GNSS-enabled monitoring highlights the growing sophistication and accessibility of WQ monitoring solutions. Innovations like smartphone-based mapping and citizen science initiatives using surfboard fins [47] further demonstrate the versatility and real-world applicability of these systems. Despite challenges, such as remote operation and data access, the integration of the IoT and LBSs has made significant progress in improving cost-effectiveness, scalability, and real-time monitoring capabilities. These advancements not only enhance the precision and efficiency of water resource management but also contribute to and emphasize future developments in environmental monitoring, ensuring the more sustainable and effective management of aquatic ecosystems. By leveraging the strengths of the IoT and LBSs, these systems empower stakeholders to make better decisions, optimize resource use, and mitigate environmental risks, ultimately contributing to the preservation and sustainable management of water resources.

Most studies focused on system functions and operations, as well as the accuracy of WQ sensors and their deployment duration before bio-fouling occurs. While some studies discussed the use of GNSSs, none evaluated the accuracy of the provided location data. The primary focus of these studies was on WQ sensor accuracy, battery consumption, and operational hours. Although some studies mentioned the type of GNSS module used, they did not assess how its accuracy affected the data collection locations or its impact on UAV operations.

This literature assessment concludes that no studies have comprehensively examined LBSs’ accuracy or evaluated their behavior in WQ monitoring. Most research focuses on system development to enhance WQ data accuracy and improve the novelty of data collection systems, such as by introducing UAVs. Additionally, efforts have been made to enhance adaptability and deployment in different water environments. However, the past literature lacks a thorough evaluation of GNSSs and LBSs, particularly in understanding how GNSS performance varies across different water bodies and surrounding environmental conditions. Based on these findings, several recommendations are proposed for future research: (i) evaluating GNSS modules alongside WQ sensors to assess their combined performance, (ii) conducting experimental studies on GNSS modules under different environmental conditions and validating them against high-accuracy GNSS data, (iii) identifying suitable GNSS modules that align with power supply requirements and environmental conditions, (iv) exploring low-power GNSS modules to enhance energy efficiency in system development, as power supply remains a critical factor, and (v) investigating GNSS behavior and accuracy variations under real-world conditions with remote and continuous monitoring, considering that GNSS modules require significant time to stabilize and acquire location data, differing from WQ sensors.

Finally, it has to be mentioned that this review was conducted using only two leading and widely recognized databases, meaning that some relevant articles may not have been included due to the selection criteria.

## Figures and Tables

**Figure 1 sensors-25-01918-f001:**
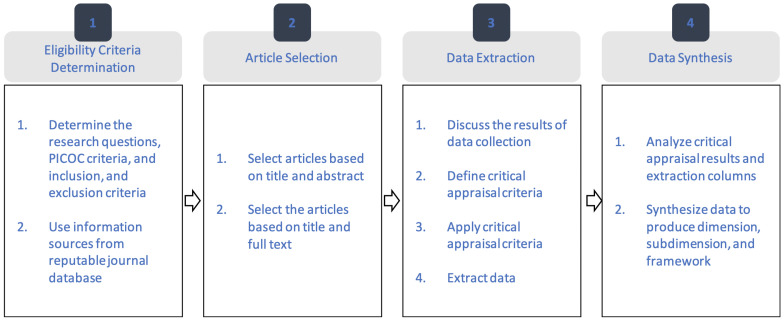
PRISMA flow chart.

**Figure 2 sensors-25-01918-f002:**
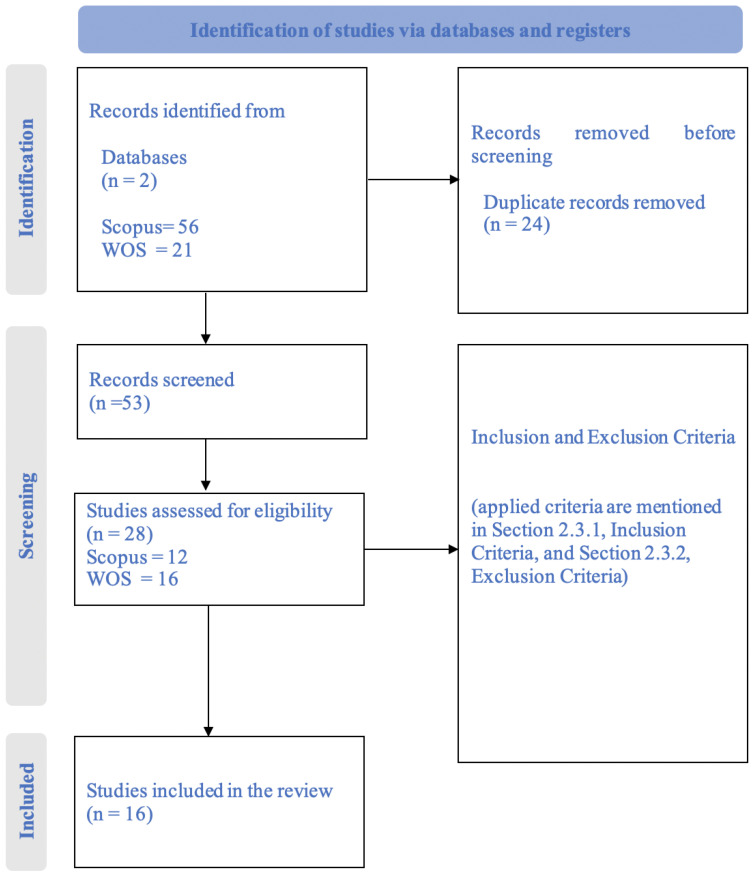
PRISMA flow diagram.

**Figure 3 sensors-25-01918-f003:**
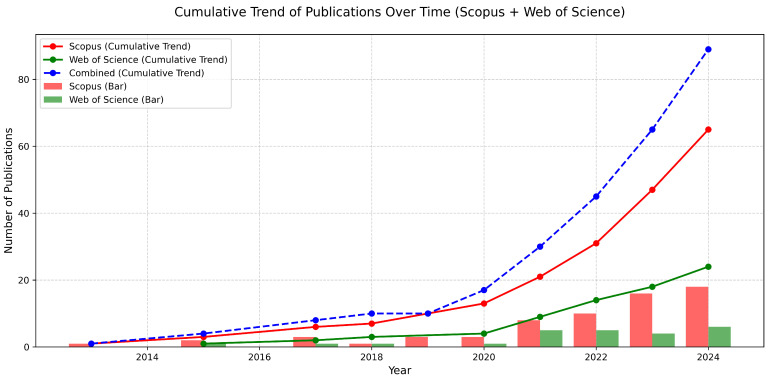
Trend of publications over time.

**Figure 4 sensors-25-01918-f004:**
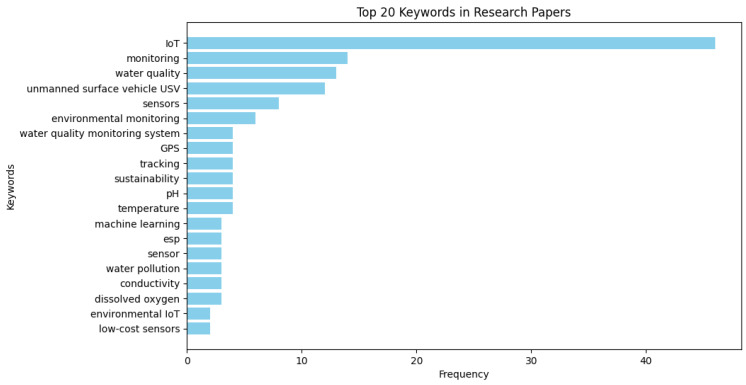
Top 20 keywords.

**Figure 5 sensors-25-01918-f005:**
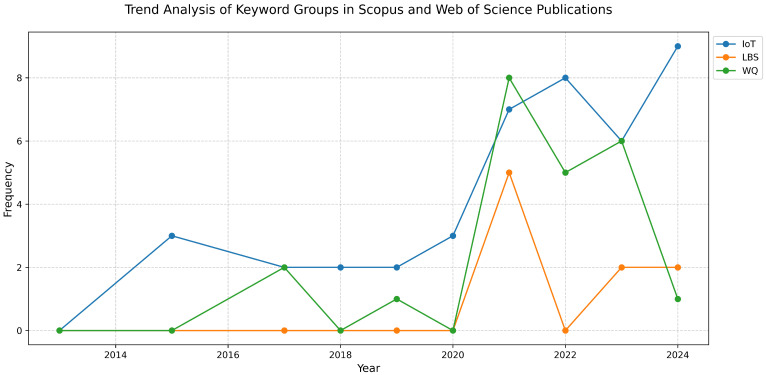
Trend analysis of keywords over time.

**Figure 6 sensors-25-01918-f006:**
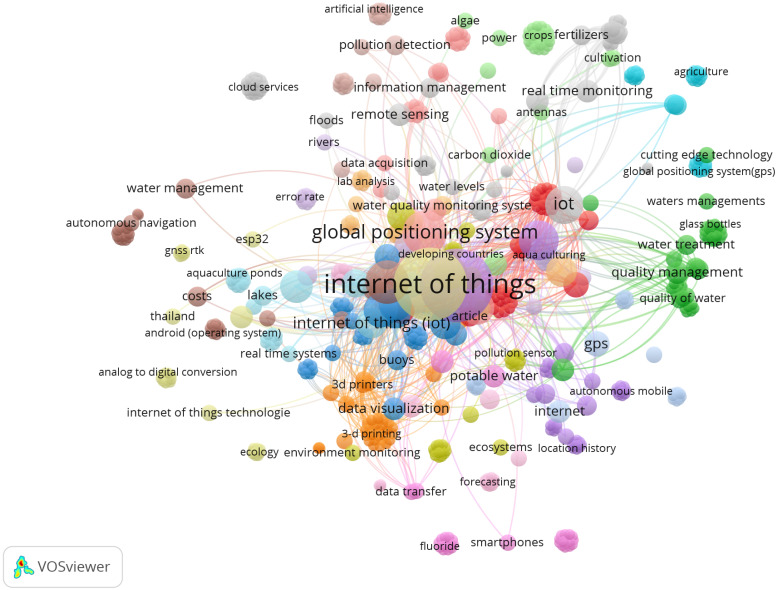
Studies by topics.

**Figure 7 sensors-25-01918-f007:**
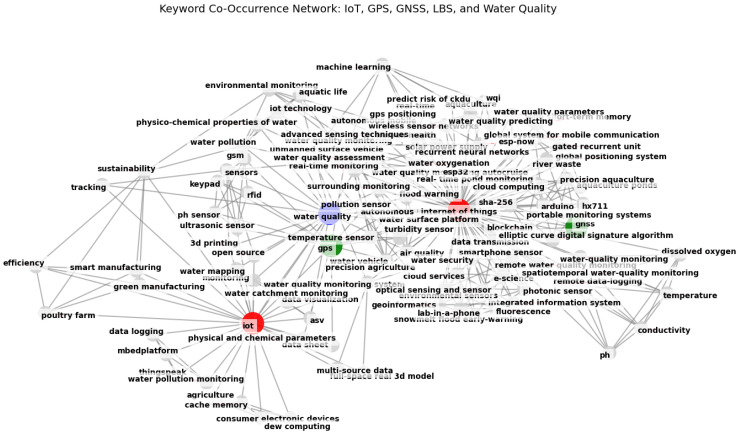
Keyword co-occurrence network.

**Figure 8 sensors-25-01918-f008:**
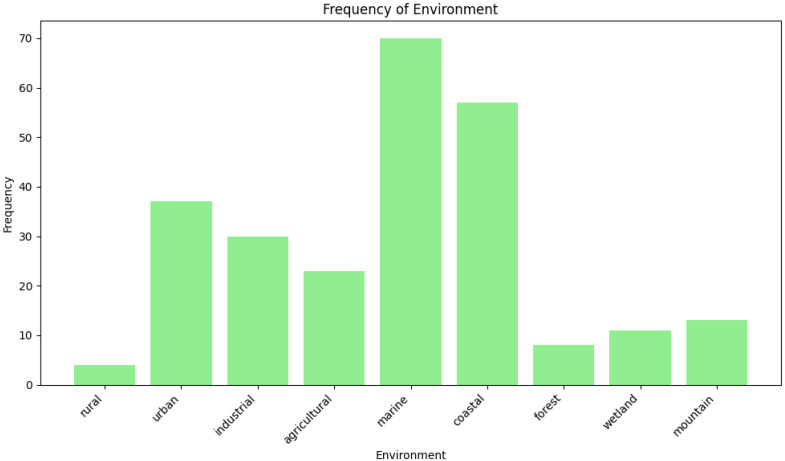
Types of environments in which the systems are deployed.

**Figure 9 sensors-25-01918-f009:**
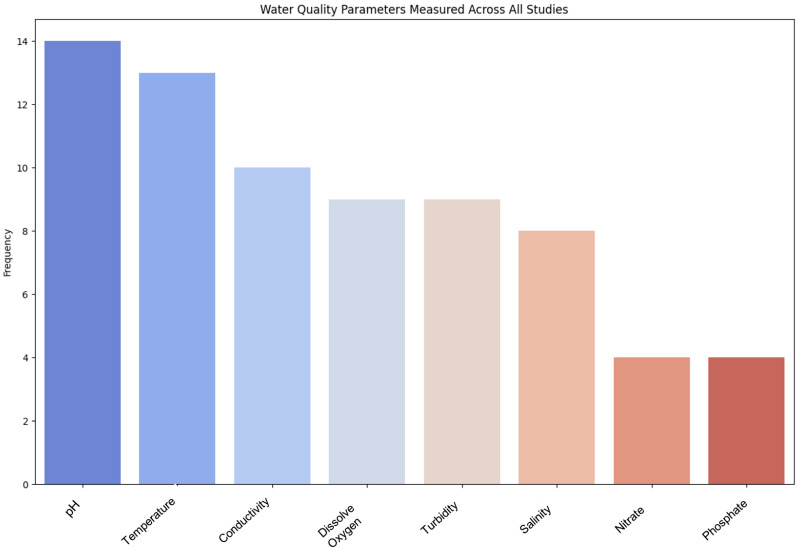
WQ parameters from all studies.

**Figure 10 sensors-25-01918-f010:**
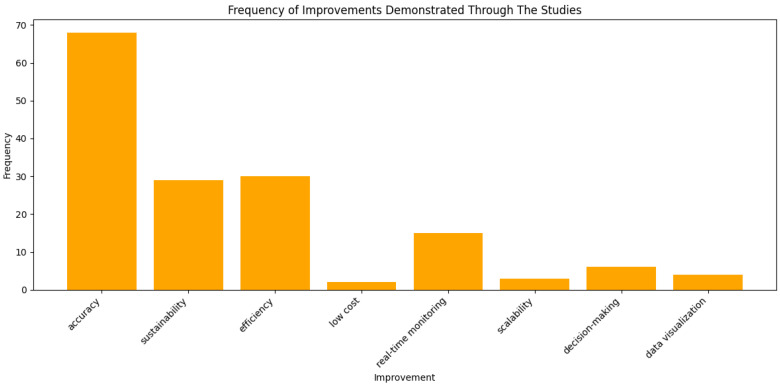
Improvements discussed in the studies.

**Figure 11 sensors-25-01918-f011:**
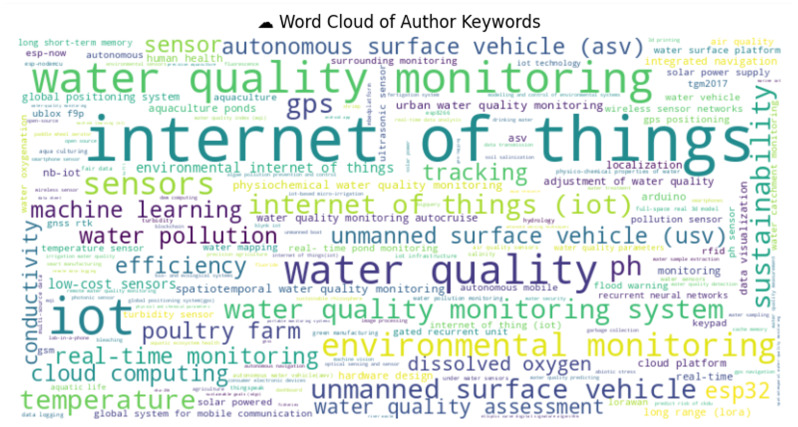
Word cloud of keywords.

**Table 1 sensors-25-01918-t001:** Overview of GNSS modules.

Device Name	Supported by GNSS Constellation	Accuracy [CEP ^1^]	Power Consumption	Communication Interface
u-blox Neo-6M	GPS	∼2.5 m	∼37 mA @ 3.3V	UART ^2^, SPI ^3^, I2C ^4^
u-blox M8N	GPS, Galileo, GLONASS, BeiDou	∼2.5 m	∼29 mA @ 3.3 V	UART, SPI, I2C
BN-180 GPS	GPS, Galileo, GLONASS, BeiDou, QZSS, SBAS ^5^	∼1–3 m	∼30 mA @ 3.3 V	UART (TTL ^6^)
u-blox CAM-M8Q	GPS, Galileo, GLONASS, BeiDou	∼2.5 m	∼27 mA @ 3.3 V	UART
LQBD1202	GPS, BeiDou	∼1 m	∼40 mA @ 3.3 V	UART
SparkFun GPS-RTK2 (ZED-F9P)	GPS, Galileo, GLONASS, BeiDou	∼1 cm (RTK ^7^)/∼1.5 m (standard)	∼68 mA @ 3.3 V	UART, USB ^8^, I2C, SPI
SIM808	GPSu	∼2.5 m	∼1 mA (sleep)/∼300 mA (active) @ 3.4–4.4 V	UART
SparkFun NEO-M8U	GPS, Galileo, GLONASS, BeiDou	∼2.5 m (with sensor fusion)	∼35 mA @ 3.3 V	UART, I2C, SPI

^1^ Circular Error Probability; ^2^ Universal Asynchronous Receiver/Transmitter; ^3^ Serial Peripheral Interface; ^4^ Inter-Integrated Circuit; ^5^ Transistor–Transistor Logic; ^6^ Satellite-Based Augmentation System; ^7^ Real-Time Kinematic GNSS Positioning; ^8^ Universal Serial Bus.

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
