# Peer review of "The Integration of IoT (Internet of Things) Sensors and Location-Based Services for Water Quality Monitoring: A Systematic Literature Review"

_sensors, 2025, doi:10.3390/s25061918_

Round 1

Reviewer 1 Report

Comments and Suggestions for Authors

This paper presents a systematic literature review on integrating IoT sensors and Location-Based Services (LBS) for water quality monitoring. It discusses recent technological advancements in these areas, their impact on real-time monitoring, and their potential to improve water resource management.

The Authors have to solve some issues according to the previous questions:

Originality and Novelty:
The topic is relevant and timely, given the increasing adoption of IoT and LBS for environmental monitoring. However, the review's novelty is limited, as many existing reviews cover IoT for water quality monitoring.
- The main contribution appears to be the integration of IoT and LBS, but this integration is not clearly emphasized as an innovation.
- The paper could be strengthened by highlighting unique insights or identifying specific gaps not covered in previous reviews.

Significance Compared to Literature:
- The review covers relevant literature but does not sufficiently compare its findings to existing surveys.
- The paper would benefit from a dedicated section comparing previous reviews and explicitly stating how this work adds value.
The bibliometric analysis is a good addition, but it does not discuss how the identified trends relate to the findings of other studies.
- The discussion should integrate more critical perspectives on limitations, biases, and unexplored areas in previous works.

Quality of Presentation
- The structure follows a standard review format, but the organization has some inconsistencies.
- The figures and tables are helpful, but some graphs lack clear labels and explanations.
- The methodology section should be more structured, possibly with a subsection dedicated to inclusion/exclusion criteria.
- Figure captions should be more detailed, explaining the significance of each graph rather than just stating what it represents.

Scientific Level
- The review is comprehensive but lacks depth in technical analysis.
- The discussion could be more analytical, critically evaluating the strengths and weaknesses of different IoT and LBS implementations.

Does the introduction provide sufficient background and include all relevant references?
- While many relevant references are included, the introduction does not provide a comprehensive state-of-the-art overview or a clear gap identification in previous literature. Some key papers on IoT-driven environmental monitoring and geospatial integration are missing.

Is the research design appropriate?
- Inclusion and exclusion criteria should be more explicit and justified.
- The selection of studies (only 28 papers) seems too narrow given the broad scope of IoT and LBS in water monitoring.
- It is unclear whether the PRISMA framework was strictly followed, as no PRISMA flowchart is provided.

Are the methods adequately described?
- The search strategy is outlined, but specific search terms, database query syntax, and filters used are not clearly detailed.

Are the results clearly presented?
- The bibliometric analysis is useful, but the results are not critically discussed; they are primarily descriptive rather than analytical.
- The figures and tables are informative, but some lack proper captions or discussions on their relevance.
- The findings from the reviewed papers are not systematically compared, making it difficult to see overarching trends or patterns.

Are the conclusions supported by the results?
- The conclusions summarize the findings, but they are mostly a restatement of the abstract rather than a synthesis of insights derived from the review.
The paper lacks a strong discussion of the implications of its findings. How should researchers or practitioners use this information?
- Future research directions are not clearly outlined, which would help strengthen the contribution.

Comments on the Quality of English Language

- The paper is generally well-written, but there are grammatical errors, awkward phrasing, and redundant expressions that reduce clarity.
Examples of errors and awkward phrasing:
"Nowadays, environmental pollutions encounter substantial challenges..." --> "Nowadays, environmental pollution presents substantial challenges..."
"The IoT provide a transformative solution..." --> "IoT provides a transformative solution..."
"To what extend these technologies enhance efficiency..." --> "To what extent these technologies enhance efficiency..."
"These advancements have made it easier to integrate location information into various applications, from urban planning and disaster management to environmental monitoring and transportation." (This sentence is overly long and could be split into two for better readability.)
The paper should undergo professional proofreading to ensure fluency and grammatical correctness.

Reviewer 2 Report

Comments and Suggestions for Authors
  • Line 78: Remove "Internet of Things" and use only "IoT" because it has already been mentioned before.

  • Decide on Consistency: Choose between using "water quality" or "WQ" consistently throughout the manuscript, as both full and abbreviated forms are currently used.

  • Figure 7: Improve the clarity of the text in the figure to enhance readability.
  • The introduction should smoothly transition into how the SLR was conducted and what specific aspects it investigates.
  • The methodology are lacks details on dataset filtering & selection criteria. 
  • While covering 2000–2024 is reasonable (Why only 5 years back), the methodology should explain why this range was chosen.
  • Conclusion needs more actionable recommendations.  No future research directions. No Discussion of Limitations.

    the other comments is in the attached file.

Round 2

Reviewer 1 Report

Comments and Suggestions for Authors

Thank you to the Authors for addressing most of my doubts and comments. However, the state-of-the-art is still poor and not completed. My recommendations about that have not been completed in total. There are works in 2024 related that have not been considered, for example, the ones I mentioned. - The review relies on some non-peer-reviewed sources, which should be replaced with more academic references. - Some key works on IoT and LBS integration are missing, and a more systematic comparison with existing reviews is needed. For example, the Authors say the limit is 2024, but the following reference is in 2024 and was not included. That is why I think the PRISMA method was not applied correctly. García Baigorri, A., Parada, R., Monzon Baeza, V., & Monzo, C. (2024). Leveraging Urban Water Distribution Systems with Smart Sensors for Sustainable Cities. Sensors, 24(22), 7223.
On the other hand, in my opinion, references from 2000 and 2004 are not relevant today.

I have one more question about the current answers: What do the authors want to express with "accuracy" in this case in the results (e.g., Figure 7)? Can they add more explanation about this?

Author Response

Dear Reviewer and Editor,

please see our response in the attacheed document. All new additions in the paper are highligthed in blue.

Many thanks and best regards,

Guenther Retscher
